# Nonsuicidal self-injury as a mediator between dissociative experiences and suicide risk in adolescents: Insights from a clinical setting

Rahime Duygu Temeltürk[1,2,3], Yusuf Gürel[4], Merve Canlı[5], Ayşegül Efe[5], Sabide Duygu Uygun[1,2,6] (iD), Fatma Hülya Çakmak[7], Miray Çetinkaya[8] and Sadettin Burak Açıkel[1,2] (iD)

[1]Department of Child and Adolescent Psychiatry, Ankara University Faculty of Medicine, Ankara, Turkey; [2]Autism Intervention and Research Center, Ankara University, Ankara, Turkey; [3]Institute of Health Sciences, Department of Interdisciplinary Neuroscience, Ankara University, Ankara, Turkey; [4]Faculty of Economics and Administrative and Social Sciences, Department of Psychology, KTO Karatay University, Konya, Turkey; [5]Department of Child and Adolescent Psychiatry, Ankara Etlik City Hospital, Ankara, Turkey; [6]Institute of Neurological Sciences and Psychiatry, Hacettepe University, Ankara, Turkey; [7]Department of Child and Adolescent Psychiatry, Beylikdüzü State Hospital, İstanbul, Turkey and [8]Private Practice, Ankara, Turkey

## Research Article

**Keywords:**
suicide; nonsuicidal self-injury; dissociation; adolescent; anxiety

**Corresponding author:**
Sadettin Burak Açıkel;
Email: acikel42@gmail.com

## Abstract

A burgeoning body of evidence suggests a higher prevalence of nonsuicidal self-injury (NSSI) behaviors among adolescents. This study aimed to examine the comorbid internalizing symptoms and suicidal behaviors, along with associations between dissociative experiences and suicide risk in adolescents attending a psychiatric outpatient unit in Ankara, Türkiye. The study included 81 adolescents aged 12–18 years, who engaged in NSSI and sought treatment at a psychiatric outpatient clinic. Psychiatric evaluations were conducted through semi-structured clinical interviews. NSSI behaviors were assessed using the Inventory of Statements About Self-Injury, and suicide risk was measured using the Suicide Probability Scale. Additionally, internalizing symptoms and dissociative experiences were evaluated using the Revised Children's Anxiety and Depression Scale-Child Version and the Adolescent Dissociative Experiences Scale, respectively. Moderate to high correlations were found among suicide risk, dissociation, NSSI severity, anxiety and internalizing scores. Mediation analysis revealed that NSSI significantly mediated the relationship between dissociation and suicide risk. These findings indicate that assessing both dissociation and NSSI could provide valuable insights into comprehending and addressing adolescent suicide, thereby facilitating the development of targeted interventions to mitigate the effects of dissociative experiences.

## Impact statement

The current study findings highlight the critical role of nonsuicidal self-injury as a mediator between dissociative experiences and suicide risk in adolescents. By identifying dissociative symptoms as a key factor in self-harming behaviors, the findings emphasize the need for systematic screening to inform targeted suicide prevention strategies. This research provides novel insights into the underlying mechanisms of suicide risk, offering valuable contributions for clinicians, researchers and policymakers working to enhance mental health interventions for at-risk youth.

## Introduction

Nonsuicidal self-injury (NSSI) refers to the deliberate and direct infliction of harm upon one's own body tissue, carried out without any suicidal intent (Nock and Favazza, 2009). This encompasses behaviors such as cutting, scratching or burning the skin, as well as striking oneself against objects, resulting in direct injury to the skin or bones (Plener et al., 2015). NSSI has become a growing focus of clinical attention, particularly among adolescents (Plener et al., 2015; Voss et al., 2020).

Several studies have investigated the prevalence of NSSI across various populations and regions, with rates ranging from ~10–35% among adolescents (Guertin et al.,2001; Çimen et al., 2017; Voss et al., 2020; Xiao et al., 2022). For instance, Voss et al. (2020) reported a lifetime prevalence of ~20% among adolescents and young adults aged 14–21 years in a study conducted in Germany. In comparison, Çimen et al. (2017) found a prevalence of ~12% among adolescents aged 12–17 years in a community sample in Turkey. In the United States, Guertin

et al. (2001) reported a prevalence of 31–33% among adolescents aged 12–18 years from a clinical population, whereas in China, Xiao et al. (2022) found a prevalence of 22–23% among adolescents from the general population. Accumulating evidence indicates that NSSI is often associated with various forms of psychopathology, including internalizing disorders such as depression and anxiety, highlighting the critical need for research on NSSI in clinical populations (Giletta et al., 2012; Bae et al., 2020; Xiao et al., 2023; Millon et al., 2024). Furthermore, the primary risks associated with NSSI are its potential to become chronic and to escalate into suicidal behaviors (Hawton et al., 2012). Notably, NSSI has been identified as a strong risk factor for suicide (Groschwitz et al., 2015).

Dissociation is often defined as an altered state of consciousness leading to reduced awareness of environmental events, which can occur both independently and in the context of psychiatric disorders (Sar, 2011). Dissociation has frequently been reported as a common concomitant symptom of NSSI, as well as of suicidal behaviors, including suicide attempts (SAs) and suicidal ideation (SI) (Sumlin et al., 2020; Bikmazer et al., 2023). Moreover, higher levels of dissociation have been identified as a significant factor in the development of NSSI and SA, regardless of psychiatric diagnosis (Kılıç et al., 2017). Given that NSSI is characterized by deliberate self-injury, whereas dissociation is typically marked by an altered state of consciousness, these two phenomena might initially seem incompatible. However, research indicates that dissociation and NSSI can indeed co-occur and are often interconnected, particularly among individuals with histories of trauma and emotional distress (Lüdtke et al., 2016; Ji et al., 2024; Sarıkaya et al., 2024). This may reflect the role of dissociation as a coping mechanism to reduce emotional pain (Ji et al., 2024) or, alternatively, the potential of dissociative symptoms, such as detachment from reality, to mediate or exacerbate NSSI behaviors (Lüdtke et al., 2016). The identification of risk factors for NSSI and suicide plays a critical role in shaping effective prevention strategies (Zalar et al., 2018). Since dissociation was found to be significantly associated with NSSI and suicidal behaviors (Bikmazer et al., 2023), clarifying the interrelationships among dissociation, NSSI and suicide plays a crucial role in prevention and early intervention. However, there is a limited body of research regarding the psychiatric evaluation of adolescents with NSSI. Moreover, the relationship between NSSI, dissociation and suicide in clinical settings remains insufficiently explored. This study addresses a gap in the existing literature by offering comprehensive psychiatric clinical data gathered through semi-structured interviews with adolescents. It investigates the co-occurrence of internalizing symptoms (anxiety and depressive symptoms) and provides detailed insights into the types and severity of NSSI. Furthermore, the study examines the mediating role of NSSI in the relationship between dissociation and suicidality.

Based on the aforementioned explanations, our primary objective was to examine the sociodemographic and clinical characteristics of adolescents with NSSI admitted to a child and adolescent psychiatry outpatient clinic in Ankara, Türkiye. To determine the clinical associations of NSSI, comorbid psychiatric disorders and internalizing symptoms, specifically anxiety and depression, were investigated. Additionally, we explored the interrelationships among NSSI, dissociation and suicide risk. Finally, we examined the mediating role of NSSI in the relationship between dissociative experiences and suicide risk, which may offer insights into treatment priorities for this population.

## Methods

### Participants and procedure

Participants were adolescents aged 12–18 years, who sought treatment for various reasons at a child and adolescent psychiatry outpatient unit in Ankara, Türkiye, between March and September 2021. Adolescents meeting Diagnostic and Statistical Manual of Mental Disorders, 5th Edition (DSM-5) criteria for NSSI – defined as intentional, self-inflicted tissue damage without suicidal intent on five or more days within the past year – and who provided informed consent were included in the study.

Exclusion criteria encompassed severe psychiatric disorders (e.g., bipolar disorder, psychotic disorders and substance use disorders), intellectual disability, specific learning disorder, autism spectrum disorder, neurological or chronic medical conditions and uncorrected visual or auditory impairments.

The research protocol was approved by the hospital's local ethics committee (ethics approval number: E-21/06–195). Verbal consent from adolescents and written consent from parents were secured before the psychiatric assessments and completion of study measures.

NSSI diagnoses were established during psychiatric assessments based on DSM-5 diagnostic criteria. Certified child and adolescent psychiatrists conducted semi-structured interviews to assess comorbid psychiatric disorders. Following this, participants completed self-report questionnaires.

Before enrollment, all participants were informed about the purpose and content of the scales, including the possibility of encountering sensitive topics. The study team closely monitored participants throughout the administration process to detect any signs of distress. Participants were informed that they could withdraw from the study at any point without consequence. In cases where the participants reported discomfort, appropriate psychological support and resources were offered to address their concerns. These procedures were implemented to prioritize the well-being of the adolescents and to minimize any potential harm. Detailed information about the clinic has been intentionally omitted to protect the confidentiality and anonymity of the participants, in accordance with ethical research guidelines.

### Measures

*Sociodemographic characteristics*, including age, medical history and family characteristics, were assessed using a semi-structured interview form. Socioeconomic status was determined with the Hollingshead–Redlich Scale (HRS) and categorized as low (≤22), medium (23–44) or high (≥45) (Hollingshead and Redlich, 2007).

*The Schedule for Affective Disorders and Schizophrenia for School-Age Children-Present and Lifetime Version* (K-SADS-PL-DSM-5), a semi-structured interview, is widely used for diagnosing child psychiatric disorders, evaluating psychiatric symptoms and ending with diagnostic supplements (Kaufman et al., 2016). The K-SADS-PL-DSM-5 Turkish version is valid and reliable (Ünal et al., 2019).

*The Inventory of Statements About Self-Injury (ISAS)* was developed to comprehensively assess the functions of NSSI (Klonsky and Glenn, 2009). This self-report measure consists of two sections: behaviors and functions. The first section, behaviors, includes 12 NSSI behaviors and asks respondents to report the frequency of each behavior over their lifetime. Additionally, it includes five items assessing the structural and descriptive characteristics of these behaviors. In the second part, the functions—referring to

the underlying motives or reasons – of NSSI behaviors are assessed through 39 items, organized under two dimensions. *Autonomous functions* comprise five subscales: affect regulation, anti-suicide, marking distress, self-punishment and anti-dissociation. *Social functions* include eight subscales: interpersonal boundaries, interpersonal influence, revenge, sensation seeking, peer bonding, toughness, autonomy and self-care. Bildik et al. (2013) conducted a psychometric examination of the ISAS in Turkey. The adapted instrument exhibited robust internal reliability, with coefficients of .79 for the first part, and .81 and .86 for the autonomous and social functions, respectively (Bildik et al., 2013). In the present study, both the behavioral and functional parts were utilized to assess the characteristics of NSSI among the study participants.

*The Revised Children's Anxiety and Depression Scale Child Version* (*RCADS-CV*) was developed to screen for internalizing symptoms in children and adolescents. This self-report questionnaire consists of 47 items and six subscales (generalized anxiety disorder, separation anxiety disorder, panic disorder, obsessive-compulsive disorder [OCD], social anxiety disorder and major depressive disorder [MDD]), as well as two composite subscales (Total Internalizing and Total Anxiety) and two comprehensive subscales (Total Internalizing and Total Anxiety) (Chorpita et al., 2000). Gormez et al. (2017) evaluated the validity and reliability of the Turkish version. Inter-scale reliability was strong with a Cronbach's $\alpha$ of .95 and subscale coefficients ranging from .75 to .86. Elevated scores correlate with heightened levels of symptoms. T scores of 65 or higher indicate scores at the borderline clinical threshold, while T scores of 70 or above denote scores exceeding the clinical threshold (Gormez et al., 2017). In the current study, the RCADS-CV was utilized to assess internalizing symptoms, specifically anxiety and depressive symptoms, among participants.

*The Adolescent Dissociative Experiences Scale* (*A-DES*) was developed by Armstrong et al. to screen for dissociative experiences in children and adolescents (Armstrong et al., 1997). This self-report scale consists of 30 items, each scored from 0 (*never*) to 10 (*always*), and is used to assess the severity of dissociative experiences. The Turkish version was found to be valid and reliable (Cronbach's $\alpha$ = .93; test–retest reliability coefficient = .91) (Zoroglu et al., 2002). In this study, the scale was used to assess participants' dissociative symptoms.

*Suicide Probability Scale* (*SPS*), developed by Cull and Gill (1982), is a 36-item Likert-type self-report scale used to evaluate suicide risk (Cull and Gill, 1982). The Turkish version was adapted, and its reliability and validity were evaluated in a clinical setting (Atlı et al., 2009). The internal consistency coefficient of the total score was reported as .87, the test–retest reliability as .98 and the concurrent validity as .84. The total score on the scale ranges from 36 to 144, with higher scores indicating an increased probability of suicide. A cutoff score of 110 has been established based on findings from a Turkish clinical sample that included both adolescents and adults (Atlı et al., 2009). In the present study, the scale was used to quantitatively assess suicide risk in adolescents with NSSI.

### Statistical analysis

IBM SPSS (Statistical Package for Social Sciences) 22.0 was used for statistical analyses of the sociodemographic and clinical characteristics of the participants. Descriptive data were presented using percentage frequency values and mean (standard deviation [SD]) values. Before the analyses, the Shapiro–Wilk test was used to determine the normality of the data distribution, and the data were found to be normally distributed. To compare suicide risk scores among groups categorized by sociodemographic and clinical characteristics, the independent samples *t*-test was employed. Pearson's correlation test was used to determine the relationships between scale scores. SPSS Process Hayes Macro model 4 was used to examine the relationship between dissociative symptoms, NSSI and suicide risk. All statistical tests were two-tailed with a threshold for significance of $\alpha$ = .05.

### Results

#### Sociodemographic and clinical characteristics of the participants

A total of 305 adolescents diagnosed with NSSI were initially recruited for the study. After applying the exclusion criteria (i.e., comorbid intellectual disorders and autism spectrum disorders), 132 participants remained eligible. Of these, 51 declined to provide consent. Ultimately, 81 adolescents with NSSI were included in the final study sample.

Of the 81 adolescents, 16 (19.8%) were male and 65 (80.2%) were female. The ages of the adolescents ranged from 13 to 18 years, with a mean age of 15.32 years (SD = 1.24). The majority of participants (80.2%) came from intact families, with a significant proportion (77.8%) originating from low socioeconomic backgrounds, as determined by the HRS (Hollingshead and Redlich, 2007). Following the administration of the K-SADS interview, psychiatric disorders were identified in all participants. Specifically, ~50% were diagnosed with MDD, one-third with attention-deficit hyperactivity disorder (ADHD), about 20% with anxiety disorders and OCD and a smaller proportion with post-traumatic stress disorder (PTSD). Tobacco use was reported by 38 participants (46.9%), alcohol use by 12 (14.8%) and cannabis use by 4 (5%), while no participants reported opioid use. Additionally, a familial history of psychopathology was documented in 36 participants (44.4%). Sociodemographic and clinical characteristics are presented in Table 1.

The mean age of NSSI onset was 12.25 ± 1.85 years. The most prevalent NSSI behaviors in the sample (*n* = 81) were cutting (*n* = 62; 76.5%), banging/hitting oneself (*n* = 48; 59.3%), interfering with wound healing (*n* = 47; 58%) and severe scratching (*n* = 39; 48.1%). The least common behavior was burning, reported by 14 adolescents (17.3%). The distribution of each NSSI behavior is illustrated in Figure 1.

Current suicidal thoughts were reported by 11 adolescents (13.6%). A history of SA was present in 34 participants (42%), with methods including medication overdose (*n* = 27; 33.3%), cutting (*n* = 4; 4.9%), hanging (*n* = 2; 2.5%) and firearm use (*n* = 1; 1.2%). Among those with a history of SA (*n* = 34), 25 (73.5%) had made a single attempt, 5 (14.7%) had made two attempts and 4 (11.8%) had engaged in three or more attempts. Eight (9.9%) participants reported a family history of suicide.

Upon examining internalizing symptoms by using RCADS-CV across the entire sample, it was found that the median scores of subscales for panic disorder and MDD, and total internalizing score, exceeded the clinical threshold (T scores: 70.5, 80 and 68.5, respectively). The total average score for dissociative experiences had a mean value of 4.5 (total score divided by the number of items), and the total score for suicide probability had a mean of 96.23 with an SD of 18.02. The total psychiatric scale scores of the participants are presented in Table 1, while detailed subscale scores are provided in Supplementary Appendix Table A1.

**Table 1.** Sociodemographic and clinical characteristics of the participants

| Sociodemographic and clinical variables | Participants (*n* = 81) Mean ± SD/*n* (%) |
|---|---|
| Sex, *n* (%) | |
| Female | 65 (80.2) |
| Male | 16 (19.8) |
| Adolescent age (years) | 15.32 ± 1.24 |
| Mothers' age (years) | 40.48 ± 5.85 |
| Fathers' age (years) | 44.43 ± 7.07 |
| Mothers' education level (years) | 6.88 ± 2.73 |
| Fathers' education level (years) | 8.07 ± 3.10 |
| Family type, *n* (%) | |
| Intact family | 65 (80.2) |
| Single parent family | 12 (14.8) |
| Extended family | 4 (4.9) |
| Socioeconomic status, *n* (%) | |
| Low | 63 (77.8) |
| Medium/high | 18 (22.2) |
| Psychiatric disorders, *n* (%) | |
| Major depressive disorder | 52 (64.2%) |
| Attention-deficit hyperactivity disorder | 30 (37%) |
| Anxiety disorder | 19 (23.5%) |
| Obsessive-compulsive disorder | 18 (22.2%) |
| Post-traumatic stress disorder | 3 (3.7%) |
| Familial psychopathology, *n* (%) | |
| Present | 36 (44.4) |
| Absent | 45 (55.6) |
| **Psychiatric scales** | **Mean ± SD** |
| Adolescent dissociative experiences scale total score | 135.09 ± 67.15 |
| Inventory statements about self-injury | |
| Autonomous functions score | 14.74 ± 7.02 |
| Social functions score | 12.39 ± 9.53 |
| Overall score | 26.85 ± 15.11 |
| Suicide Probability Scale total score | 96.23 ± 18.02 |
| Revised child anxiety depression scale-child version | |
| Generalized anxiety disorder | 56.98 ± 11.95 |
| Separation anxiety disorder | 59.52 ± 12.84 |
| Panic disorder | 66.98 ± 13.39 |
| Obsessive-compulsive disorder | 62.14 ± 10.67 |
| Social phobia | 50.48 ± 14.78 |
| Major depressive disorder | 80.65 ± 16.59 |
| Total anxiety | 60.70 ± 14.42 |
| Total internalizing | 66.59 ± 11.46 |

*Note:* Socioeconomic status was assessed using the Hollingshead–Redlich Scale: low (≤22), medium (23–44) and high (≥45).SD, standard deviation.

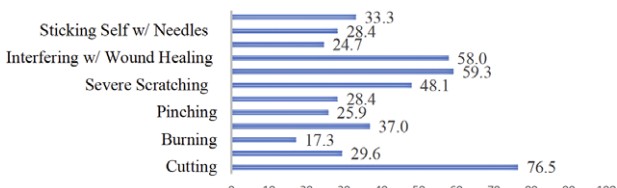

**Figure 1.** Distributions of nonsuicidal self-injury behaviors among the adolescents.

When comparing SPS scores across groups differentiated by sociodemographic and clinical characteristics, adolescents from single-parent families demonstrated significantly higher SPS scores compared to their counterparts (independent samples *t*-test, *t* = 3.08, *p* = .003). Higher scores were also observed among those reporting current SI and engaging in NSSI behaviors such as carving and burning. Specifically, SPS scores were significantly elevated in adolescents from single-parent families and those with current SI (independent samples *t*-tests, *t* = 3.08, *p* = 0.003 and *t* = 3.12, *p* = 0.003, respectively) (see Table 2).

Correlation analysis among psychiatric scale scores in adolescents revealed moderate to high correlations (0.30–0.50 and >0.50, respectively) (Cohen, 1988) were observed among the total scores of the SPS, A-DES and ISAS (social and autonomous), as well as the RCADS-CV total anxiety and internalizing scores (see Table 3).

To further determine the interrelationship between dissociative experiences, NSSI and suicide risk, we used the Process Hayes Macro model 4, which was developed based on the hypothesis that NSSI could mediate the relationship between dissociation and suicidal probability. RCADS-CV depression score was included as a covariate in this model. The mediation model posits that dissociative experiences may lead to NSSI, which, in turn, increases suicide risk, thereby establishing both direct and indirect pathways between dissociation and suicide risk. The proposed model demonstrated statistical significance (*p* = .001; $R^2$ = .36). Finally, the significant correlations of both dissociative experiences and NSSI (ISAS-total score) with SPS (*r* = 0.47, *p* < .001 and *r* = 0.53, *p* < .001, respectively) suggested a partial mediating effect of NSSI on the relationship between dissociative experiences and suicidal risk score. RCADS-CV depression subscale score did not have a significant effect on NSSI (*b* = .020, standard error [SE] = .014, *t* = 1.43, *p* = .158) or on SPS scores (*b* = .008, SE = .016, *t* = .48, *p* = .634) (see Table 4).

## Discussion

The present study examined internalizing symptoms, dissociative experiences and suicide probability among adolescents diagnosed with NSSI who attended a psychiatric outpatient clinic of a secondary-level training and research hospital in Ankara, Türkiye. As a major metropolitan city, Ankara serves a socioeconomically diverse population; however, most referrals to the clinic were from adolescents in lower socioeconomic backgrounds, often with limited access to community mental health services and ongoing family-related stress. These contextual factors likely influenced the severity and presentation of psychiatric symptoms, including non-suicidal self-injury and internalizing disorders.

**Table 2.** Comparison of suicide probability scale scores across groups

| Variables | Suicide probability scale Mean ± SD | *t* | *p* |
|---|---|---|---|
| **Sex** | | | |
| Female (*n* = 65) | 97.65 ± 17.55 | 1.43 | .157 |
| Male (*n* = 16) | 90.50 ± 19.33 | | |
| **Family type** | | | |
| Intact and extended family (*n* = 69) | 93.91 ± 16.50 | 3.08 | .003* |
| Single-parent family (*n* = 12) | 111 ± 20.97 | | |
| **MDD diagnosis** | | | |
| Absent (*n* = 29) | 92.06 ± 17.98 | 1.57 | .121 |
| Present (*n* = 52) | 98.56 ± 17.79 | | |
| **OCD diagnosis** | | | |
| Absent (*n* = 63) | 95.90 ± 18.37 | 0.31 | .760 |
| Present (*n* = 18) | 97.39 ± 17.18 | | |
| **Socioeconomic status** | | | |
| Low (*n* = 63) | 97.22 ± 17.79 | −0.92 | .359 |
| Medium–high (*n* = 18) | 92.78 ± 18.89 | | |
| **Current suicidal ideation** | | | |
| Absent (*n* = 70) | 93.89 ± 17.07 | 3.12 | .003* |
| Present (*n* = 11) | 111.18 ± 17.39 | | |
| **History of suicidal attempt** | | | |
| Absent (*n* = 65) | 94.68 ± 15.44 | 0.91 | .365 |
| Present (*n* = 65) | 98.38 ± 21.13 | | |
| **NSSI/burning** | | | |
| Absent (*n* = 67) | 94.81 ± 18.77 | 1.58 | .048** |
| Present (*n* = 14) | 103.07 ± 12.21 | | |
| **NSSI/carving** | | | |
| Absent (*n* = 51) | 91.92 ± 16.36 | 2.94 | .004** |
| Present (*n* = 30) | 103.57 ± 18.60 | | |

*Note:* Independent samples *t*-test. MDD, major depressive disorder; NSSI, nonsuicidal self-injury; OCD, obsessive-compulsive disorder; SD, standard deviation.
*$p$ < .01,
**$p$ < .05.

**Table 3.** Correlations between psychiatric scale scores of participants

| | A-DES | ISAS-Autonomous | ISAS-Social | SPS | RCADS-Total anxiety |
|---|---|---|---|---|---|
| ISAS-Autonomous | 0.52*** | | | | |
| ISAS-Social | 0.43*** | 0.59*** | | | |
| SPS | 0.47*** | 0.65*** | 0.39*** | | |
| RCADS-Total anxiety | 0.41*** | 0.29* | 0.32** | 0.32** | |
| RCADS-Internalizing | 0.48*** | 0.36** | 0.33** | 0.49*** | 0.89*** |

*Note:* Pearson's correlation test. A-DES, Adolescent Dissociative Experiences Scale; ISAS, inventory statements about self-injury; RCADS, Revised Child Anxiety Depression Scale; SPS, Suicide Probability Scale.
*$p$ < .05,
**$p$ < .01,
***$p$ < .001.

**Table 4.** Outcomes of mediation analysis

| | *b* | *SE* | *t* | *p* |
|---|---|---|---|---|
| | | **Direct and total effects** | | |
| ADES ⟶ SPS | .08 | .03 | 2.78 | .007 |
| ADES ⟶ ISAS | .12 | .02 | 5.09 | <.001 |
| ISAS ⟶ SPS | 0.47 | .13 | 3.72 | <.001 |
| | | **Bootstrap results for the indirect effect** | | |
| | Boot *b* | SE | Boot LLCI | Boot ULCI |
| Effect | .20 | .06 | .07 | .32 |

*Note*: ADES, Adolescent Dissociative Experiences Scale; *b*, unstandardized regression estimate; *β*, standardized regression estimate; CI (lower), confidence interval lower bound; CI (higher), confidence interval higher bound; ISAS, inventory statements about self-injury; SE, standard error of unstandardized estimate; SPS, Suicide Probability Scale.

Wilkinson, 2016), as females are more likely to engage in self-cutting than males. Similar to the previous literature, there is a predominance of female participants in our sample. The higher prevalence of cutting behavior among females could lead to the overrepresentation of cutting in this population (Cipriano et al., 2017).

It was previously established that depression and anxiety disorders are risk factors for NSSI (Giletta et al., 2012; Xiao et al., 2023; Millon et al., 2024). Accordingly, depression and anxiety symptoms, ADHD, OCD and PTSD were found to be more common in adolescents who exhibited NSSI behaviors (Çimen et al., 2017; Cipriano et al., 2017; Balázs et al., 2018; Sarıkaya et al., 2024). In line with the previous literature, we found that over half of the adolescents had depression; approximately one-third had ADHD. Additionally, anxiety and OCD were diagnosed in approximately one-fifth (22.2%) of the sample, whereas only 3% had PTSD. The relatively low incidence of PTSD may be attributed to the fact that trauma victims (i.e., childhood maltreatment and sexual abuse) have not been included in studies, as research has generally focused on adolescents (Cipriano et al., 2017; Steine et al., 2020; Bikmazer et al., 2023).

The current study revealed that participants' total dissociation score was ~4.5. This finding is consistent with the results of the

Cutting was the most prevalent NSSI behavior, particularly among females. More than half of the participants were diagnosed with MDD, alongside notable rates of ADHD and anxiety disorders. Dissociation and suicide risk scores were consistent with prior clinical findings, underscoring the serious risk of suicidality associated with NSSI. Correlational analyses revealed strong associations between dissociation, NSSI severity and suicide risk, while mediation analysis showed that NSSI partially mediated the relationship between dissociation and suicide probability.

According to our results, cutting was the most prevalent form of NSSI behavior among adolescents, which is consistent with the prior research (Hawton, 2004; Swenson et al., 2008). Additionally, banging or hitting oneself was the next most common NSSI type, similar to other studies (Çimen et al., 2017). Some studies suggest that self-injury is commonly associated with cutting (Cassels and

Turkish validity and reliability study of the A-DES (Zoroglu et al., 2002), which reported dissociation scores ranging from ~2 to 4 among adolescents in a clinical sample, including individuals diagnosed with PTSD, anxiety disorders, mood disorders and ADHD. Notably, the psychiatric disorders observed in the previous study align closely with the diagnostic profiles typically seen in our clinical setting. Additionally, in our study conducted in a clinical setting with adolescents exhibiting NSSI behaviors, the mean suicide risk score of adolescent participants was around 95. Although the score falls below the established cutoff score for suicide risk (110), it nonetheless suggests a clinically significant level of suicidality and warrants careful consideration (Atlı et al., 2009). While NSSI behaviors are not necessarily indicative of explicit suicidal intent, they constitute a substantial risk factor for suicidality. In light of the established limitations of screening instruments in reliably predicting suicidal behavior, the current findings should be interpreted cautiously and viewed as exploratory.

It is well-documented that NSSI is strongly associated with suicidal behavior (Asarnow et al., 2011; Hamza et al., 2012; Poudel et al., 2022). Likewise, in the present study, moderate-to-high correlations were detected between SPS and ISAS social and autonomous scores. Similarly, the observed relationships between NSSI and dissociation are consistent with a recent study that demonstrated higher dissociative symptoms in adolescents with NSSI compared to both clinical and nonclinical control groups without NSSI (Sarıkaya et al., 2024).

Prior research indicated that the frequency and severity of NSSI and the number of NSSI methods were associated with SA (Wang et al., 2022). In accordance with the literature, our findings revealed that adolescents presenting with NSSI-burning were more likely to exhibit higher SPS scores. Additionally, in line with previous findings indicating that low parental support and high familial problems, including dysfunction and/or conflict, are potential risk factors for suicide (Asarnow et al., 2011; Wilkinson et al., 2011; Hamza et al., 2012), our findings also reveal that children of divorced parents exhibited a heightened risk of suicide. Additionally, we found significant associations between suicide probability and both anxiety and depressive symptoms, which supports existing studies that identify depression as a key predictor of suicidal behavior (Asarnow et al., 2011; Poudel et al., 2022).

Based on our research findings, dissociation is directly associated with both NSSI and suicide, and additionally exerts an indirect effect on suicide, with NSSI mediating this effect. Although depressive symptom severity was incorporated into the mediating model, the mediating effect of dissociation remained unchanged. This finding is consistent with recent research suggesting that dissociative symptoms may serve as predictive factors for NSSI in adolescents with a history of childhood sexual abuse (Bikmazer et al., 2023). Although our study did not primarily include trauma victims, the mediation results were still similar, despite this significant difference in the study population. Another recent study of clinically referred early adolescents (ages 11–13 years) found that daily dissociative experiences were linked to elevated suicide risk (Vine et al., 2020). Similarly, a recent longitudinal study showed a correlation between the presence of severe dissociative symptoms in the general adolescent population and an increased risk of NSSI in follow-up, emphasizing that intensive attention should be given to such a specialized group (Tanaka et al., 2024). Nonetheless, given the complex and multifactorial nature of suicidality, further longitudinal and experimental studies are required to clarify the temporal and causal relationships between dissociation, NSSI and suicide risk.

From these results, we can infer that, as previously indicated in the literature (Steine et al., 2020; Sumlin et al., 2020), dissociative symptoms emerge as critical factors to consider when assessing the risk of suicidal behavior and NSSI among adolescents in clinical populations. These findings emphasize the need for a comprehensive evaluation of dissociative symptoms in adolescents, as they play a pivotal role in both NSSI and suicidality. However, caution is warranted in interpreting these results as dissociation may be a contributing, rather than a determinative, factor. Our findings should be regarded as preliminary and hypothesis-generating, emphasizing the importance of integrating dissociation assessment into broader clinical evaluations without overestimating its predictive capacity.

### Strengths and limitations

A notable strength of this study is its exclusive focus on adolescents drawn from a clinical population, providing a solid basis for drawing meaningful conclusions about this vulnerable group. Additionally, the incorporation of psychiatric interviews alongside self-report scales enhances the accuracy of the findings by minimizing measurement bias. Third, both dissociation and NSSI have been suggested to be associated with childhood trauma, whereas the current study was conducted on a sample largely composed of individuals who had not experienced life-threatening events. Last, despite the cross-sectional design of the study, the mediation analysis provided evidence – albeit limited – supporting the theoretical relationships between dissociation, NSSI and suicidal behavior.

However, several potential limitations should be acknowledged. First, the cross-sectional design limits the ability to infer causal or longitudinal relationships. Second, the reliance on a clinical sample of adolescents with psychiatric disorders constrains the generalizability of the findings. Furthermore, the clinical nature of the sample introduces heterogeneity, as some participants were receiving medical treatment for their psychiatric conditions. Future research is warranted to examine the longitudinal relationship between dissociation, NSSI and suicidality over a defined and shorter timeframe. Additionally, examining the potential effects of various therapeutic interventions on NSSI and its associated behaviors, particularly in adolescents, represents an important direction for future research. Longitudinal research tracking changes over time could help clarify the directionality of the relationships between dissociation, NSSI and suicidality. Moreover, incorporating diverse clinical and community-based samples may provide a more comprehensive understanding of these behaviors across different settings and populations. Research should be expanded to include diverse cultures and ethnicities to better understand how cultural factors shape these behaviors. NSSI may be considered a disorder in its own right; therefore, establishing a clear clinical definition and developing a standardized assessment tool are crucial. Such steps would strengthen research, facilitate longitudinal and cross-cultural studies and enhance our understanding across diverse ethnic contexts, although further work is still required.

### Conclusion

The implications of this study highlight the importance of considering dissociative symptoms when assessing adolescents at risk for NSSI and suicide. The direct and indirect relationships between

dissociation, NSSI and suicide risk underscore the need for early interventions targeting dissociative experiences in clinical settings. Given the association between dissociation and both NSSI and suicidality, clinicians should be particularly vigilant in identifying and addressing dissociative symptoms in adolescents, as these may serve as key indicators of heightened risk. Furthermore, the findings suggest that a comprehensive, multifaceted approach – including the evaluation of dissociative experiences alongside other psychiatric symptoms – may enhance suicide prevention and intervention strategies for this population.

**Open peer review.** To view the open peer review materials for this article, please visit http://doi.org/10.1017/gmh.2025.10079.

**Supplementary material.** The supplementary material for this article can be found at http://doi.org/10.1017/gmh.2025.10079.

**Data availability statement.** The data supporting these findings are not publicly accessible due to ethical restrictions, but can be obtained upon reasonable request from the corresponding author.

**Acknowledgments.** The authors thank all the clinic team involved with data collection. The authors would also like to express their sincere appreciation to Gökçe Yağmur Efendi for her meticulous editorial assistance during the final stages of this manuscript.

**Author contribution.** RDT: Conceptualization, methodology, data curation, analysis and writing original draft preparation. SBA: Conceptualization, methodology, data curation, reviewing and editing. YG: Conceptualization, methodology, data curation, reviewing and editing. MC: Conceptualization, methodology and data curation. AE: Conceptualization, methodology and data curation. SDU: Conceptualization, methodology and data curation. HÇ: Conceptualization, methodology and data curation. MÇ: Conceptualization, methodology and data curation.

**Financial support.** This research did not receive any specific grant from funding agencies in the public, commercial or not-for-profit sectors.

**Competing interests.** The authors declare none.

**Consent to participate.** Verbal consent from adolescents and written consent from parents were secured before the psychiatric assessments and completion of study measures. This study adheres to the principles outlined in the Declaration of Helsinki.

**Ethics statement.** This study was performed in accordance with the ethical standards as laid down in the 1964 Declaration of Helsinki and its later amendments or comparable ethical standards. Written informed consent was obtained for all participants before the initial assessment. Ethical approval was granted through the human research ethics committee of Ankara Dr. Sami Ulus Training and Research Hospital for Maternity and Children's Health and Diseases (*Approval Number:* E-21/06–195).

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
