## [Reviewer Report]

The manuscript addresses a highly significant and timely topic by exploring the mediating role of nonsuicidal self-injury in the relationship between dissociative experiences and suicide risk among the adolescent age group. The study has the potential to make meaningful contributions to the literature on interventions in nonsuicidal self-injury group. However, certain methodological details and the discussion section could benefit from further clarity and depth to enhance the overall impact of the paper. Suggestions for improving the quality of this manuscript were given below.

First of all, certain sentences and phrases are ambiguous or overly complex, potentially making the content difficult to understand. A thorough review of the manuscript’s English language is recommended to enhance clarity, grammatical accuracy, and coherence.

Introduction:

Including conceptual explanations of the variables (dissociation, nonsuicidal self-injury and suicide) assessed within the scope of the research in this section is a commendable approach. However, the introduction section of the manuscript appears to lack coherence and organization in terms of logical flow and overall structure. This impacts the clarity of the narrative and makes it challenging for readers to follow the context and objectives of the study. A more structured and cohesive approach is recommended to ensure a clear connection between the background information, the research gap, and the aims of the current study.

Methods:

Providing a more comprehensive explanation of the inclusion and exclusion criteria is recommended. Furthermore, incorporating numerical data on the number of adolescents included in the study, along with detailed reasons for their exclusion, would enhance the clarity and rigor of the methodology.

It is obvious that the scales have been thoroughly and comprehensively described. However,clarification is needed regarding the method and setting in which the scales were administered. Additionally, in this study conducted within outpatient clinic conditions, it would be important to address whether the completion of these scales had any adverse effects. Providing a more detailed explanation on these aspects would enhance the methodological transparency.

Results:

In the Results section, the tables and figures have been presented in an organized manner, accompanied by detailed explanations.

Discussion:

Although the findings have been discussed in light of the hypotheses, the practical implications of the findings could be further elaborated, especially in terms of psychosocial support programs for this group. The discussion should also incorporate suggestions for future research directions to offer a broader perspective on the study’s findings.

---

## [Reviewer Report]

Overall comments:

Overall, the manuscript demonstrates significant deficiencies in coherence and focus. The objectives are misaligned with the study title, and the content lacks logical progression, particularly with the abrupt inclusion of OCD, anxiety, and associations with psychopathologies in an unrelated context. Furthermore, the citations provided do not sufficiently verify the statements made, which is a critical flaw in scientific writing. Additionally, the study lacks novelty, significance, and a compelling justification for its conduction, further undermining its contribution to the field. The methodology section failed to elucidate its core components. The manuscript is not ready for publication in its current state.

Title of the study: I understand the meaning of “Clinical sample” term, but it can be misleading and disrespectful to readers and participants of this study. Since psychology works with living human beings, it is better to treat research participants as human beings. I suggest using alternative words against the term “clinical sample”.

Impact statement: The impact statement explains the results of this study rather than highlighting the study’s impact. Could you please revamp it to emphasize this study’s impact and significance?

Introduction:

In lines 17-20, the authors have provided the prevalence range (10% to 45%) of NSSI among adolescents where they extracted information from three articles that were published in 2017, 2016, and 2020. Does this prevalence range the finding for the cited three articles, or did the authors calculate it by themselves? I have gone through your cited article written by Çimen et al. (2017) and Voss et. Al. (2020) and for your information they did not calculate any prevalence of NSSI, rather they cited it. Please go check your articles again. In this section, I suggest including additional information such as the region and nature of the participants.

In lines 22-29, the article you cited does not support your claimed statement. The study by Bae et al. (2020) states that “patients with NSSI and suicide attempts were more likely to suffer from Cluster B personality disorder than the other groups.” However, after thoroughly reviewing the results section of this study, I did not find any information that allies with your statement.

In lines 38-43, the definition of dissociation states that it involves an altered state of consciousness. Could you please clarify how dissociation can be an important factor in the NSSI study, considering that NSSI includes intentional self-injurious behaviors, while dissociation often involves an altered state of consciousness? Do these two concepts contradict each other? Please explain how they can coexist within the context of your research.

In lines 45-50, here is the excerpt from the article of Kılıç et al. (2017): “However higher level dissociation seems as an important mediating factor, even regardless of psychiatric diagnosis, in the development of SIB and SA.” My first concern is the authors have copied almost the entire text from this article. The second concern is that “announced” and “seems” do not convey the same meaning. Additionally, the factor and mediating factor are not the same, and “suicide” and “suicide attempt” are not the same. Please read the cited articles carefully and paraphrase them well without distorting the meaning of them.

However, this section does not articulate the relationship between dissociation experience and NSSI concisely, because self-injury behaviors (SIB) and NSSI have distinguished differences that the authors did not express. They tried to establish a relationship between dissociation and SIB, and SIB appears here unexpectedly. Hence, coherence is needed. Please revamp this section.

In lines 11-30 (page no 5), the objectives of this study do not align with the title of this study. Firstly, it is not clear why you need to investigate the internalizing problems of the respondents. Secondly, why do you need to determine the characteristics of NSSI instead of exploring mediating role of it in the relationship between dissociative experiences and suicide risk? Thirdly, you used the term suicide probability where the study focuses on suicide risks. Fourthly, you mention OCD, anxiety, and depressive symptoms without any clear connection to the study. What is the justification for including these topics in the context of your research? Finally, you have mentioned a gap, but you did not state about this gap. You should address the gap in the earlier section of this introduction. Please set the objectives that align with the title of study.

Method: Materials are built with the methodology section. Methodology or Method will be the appropriate term instead of materials and methods. Please include the research design, sampling technique, and data collection procedure in the method sections. A dedicated section would be better to explain them thoroughly. Please address the following queries in the appropriate sections:

1. It is unclear how participants were recruited. Why did you choose to include all patients seeking medical treatment in your study setting? How were your inclusion criteria determined? Considering that the participants were recruited from clinical settings, they may have exhibited varying levels of psychological issues, including potential psychotic disorders. In light of this, please elaborate on the exclusion criteria employed to address this concern.

2. Dissociative experiences are one of the core topics for this study, you have applied A-DES scale to screen dissociative experiences among participants. Is there any role of this scale in recruiting participants for this study? If not, if any participants did not experience any dissociative experience, were they excluded from the study? If yes, how were the participants able to respond to these scales?

3. I noticed that you have applied several psychometric scales, but it is unclear how these scales were administered. Who administered these scales? How did you comply with ethical standards (e.g., if participants feel distress or recall any traumatic memory, or confidentiality issues) during its application?

4. In line 10 (page 6), ISAS was used to evaluate SIB of participants. SIB includes both non-suicidal and suicidal behavior. However, in the description of ISAS (line 45), you stated that this scale measures NSSI. Please be consistent with the accurate information.

5. You have employed SPS, but your study is about suicide risk. How is this scale related to this study?

6. Whenever you will state scale name, please include citation. It is very difficult to extract information through the name of the scale only.

In line 22 (page 8), SPSS stands for Statistical Package for Social Sciences. Please write package instead of program. Please include the reason for running an independent sample t-test.

Results:

I did not find any report addressing dissociative experiences among the participants. Additionally, the results section does not align with the stated objectives of this study. For example, where is the analysis of co-occurrence outcomes for internalizing symptoms?

In lines 6-10 (page 13), a hypothesis is mentioned for the first time in the results section. This hypothesis should have been introduced earlier in the introduction section.

I suggest revisiting and refining the objectives of the study. Based on the revised objectives, it would be appropriate to run the analyses again to ensure that the results address the study’s aims comprehensively.

Discussion:

In line 38 (page 14), the study findings indicate that one-third of respondents were diagnosed with ADHD. Could you explain how you administered the scales to these participants, given that this study employed several scales comprising a plethora of questionnaires? Participants with ADHD may face challenges in responding accurately due to attentional difficulties. Please elaborate on how you addressed these challenges in the scales application section.

In lines 52-54, please include the exact correlation score and clarify the criteria you used to determine whether the correlation was moderate or high. In lines 4-10 (page 15), the logic presented is unclear; please articulate it more concisely for better comprehension.

In lines 10-15, you cited that “low parental support and high family conflict are potential risk factors for suicide (Hamza et al., 2012).” Could you clarify how children of divorced parents might experience high family conflict? Since your data supports this finding, it is crucial to provide a clear explanation of the logic behind this claim.

In lines 24-36, you assert that your findings are consistent with those of Bikamazer et al. (2023). However, have you collected data on childhood sexual abuse history? If not, this claim lacks validity and should be reconsidered.

Finally, my recommendation will be to revamp the discussion section according to your revised study objectives and results.

---

## [Reviewer Report]

The Mediating Role of Non-suicidal Self-Injury in the Relationship Between Dissociative Experiences and Suicide Risk Among Adolescents: Insights from a Clinical Sample

The authors are examining the comorbid internalizing symptoms and suicidal behaviors in a clinical sample of adolescents. In addition, their effort is to understand associations between dissociative experiences. NSSI and suicidal probability.

They conclude (among other conclusions) that increased probability of suicide in those who have dissociative experiences is mediated through NSSI.

Impact statement:

NSSI abbreviation- please make sure that the abbreviation is listed the first time in the full form and later used consistently in the manuscript.

This should include a statement about the reason for conducting the study and the implications of the findings in the clinical practice and real world rather than listing the findings again.

Introduction: overall comments

Background literature around association of NSSI, suicide and dissociation in young people needs to be stated clearly. How this literature has informed the research questions needs to be clarified.

Introduction lines 15-18

Kindly state prevalence rates in adolescence with adolescent specific references.

Lines 31-32

What other self-injurious behaviours are the authors referring to (other than suicide attempts). Also, if authors are differentiation NSSI from suicide attempts, kindly elaborate in a statement how the two differ.

Lines 45-47

Higher level dissociation has been announced as an important factor, even regardless of psychiatric…

“of” is missing. Kindly make the correction.

Authors introduce self-injurious behaviour in this line- how is this different to NSSI.

Avoid using multiple terminology to suggest the same phenomena, any new term need to be described and differentiated from those used earlier.

Page 4, lines 13-16

…who admitted to the child and adolescent psychiatry outpatient clinic.

Lines 21-23

…dissociation and suicide probability, which might provide to gain insight into what should be focused on in terms

Kindly correct the Grammar.

Page 4, lines 11-30

This section suggests 3 aims of the study, these are not reflected clearly in the abstract. Also, the population under investigation is restricted to those admitted in a child and adolescent unit which needs to be specified in the abstract.

Page 4, lines 38-40

The sample consisted of 81 adolescents (13-18 years old), with a mean age of 15.32 years (SD = 1.24), who applied to the child and adolescent psychiatry outpatient clinic of ….. Hospital.

Kindly clarify what does applying to outpatient clinic mean.

Also, please remove any gaps such as… and replace it with information needed to understand the context.

Page 5, lines 3-19

Kindly remove the information about assessment from participants section and add to measures.

Table 1. Psychopathology in participants need to be specified (along with the gender distribution of the psychopathology, if possible)

Results:

The mediation models need to be explained a bit more for the readers to understand the associations.

Discussion:

This section needs more development. The authors need to list the crucial findings initially. and help the readers understand the meaning and the practical importance of each in subsequent paragraphs. Of critical importance is to list the implications of the findings.

Page 14, line 8

…who presented NSSI/ burning was…

The sentence doesn’t read right. Kindly correct it to add with. Also, make the same correction wherever else it features in the manuscript.

…presented with NSSI…

Conclusion:

The conclusion consists of a set of general statements about NSSI in adolescents. Kindly make it specific derived from the findings of the current study.

---

## [Editor Report]

Dear Dr Açıkel,

We have received three reviews for your article from experts in the field. Reviewers two and three collectively outlined key areas of improvement needed for the article to be ready for publication. I encourage you and the team to read the comments carefully and to address each one.

Thank you and all the best,

Dr. Sandersan Onie

---

## [Reviewer Report]

I have reviewed the revised version of the manuscript, which has been revised in accordance with the reviewers' suggestions. The authors have carefully implemented the recommended revisions, and the scientific quality of the manuscript has been enhanced.

In its current form, I find the manuscript suitable for publication.

---

## [Reviewer Report]

Thank you for your thorough revisions, which have significantly strengthened the manuscript. I am pleased to share that it has been accepted for publication.

As a final check from my side, I kindly request that the keywords on page 3 be made consistent with those listed on the first page.

---

## [Editor Report]

Dear Prof Açıkel, 

Thank you for your resubmission. As you can see, the reviewers are both satisfied with the revisions - as am I. Before acceptance, I just have one final point:

’A notable strength of this study is its exclusive focus on a clinical population; by examining adolescents in a clinical setting, the study directly addresses the inherent vulnerabilities of this age group, thereby enhancing its overall relevance.'

This sentence here is a little vague. Does the strength come from examining adolescents in a clinical setting, or does it come from examining adolescents from a clinical population, which is allows us to reach conclusions on this vulnerable group? Please revise for clarity. 

All the best,

Dr. Sandersan Onie

---

## [Editor Report]

Dear Prof Açıkel,

I am now satisfied with the revision and will recommend this article for publication.

Thank you and all the best,

Sandersan Onie

---

## [Editor Report]

Dear Prof Burak and colleagues,

Thank you for thoroughly revising the manuscript. Upon re-reading, I have some more comments that may help the reader understand the important work being done in this study:

1. It would be good to provide more information on the scales e.g., the SPS would benefit from having a score range outlined, with cutoffs to help readers who are not familiar with it make sense of the scores. For example, in the discussion, it was stated that the score was 95, on page 14 - without context this offers little meaning to the reader. 

2. Suicide is a very challenging topic to measure, especially when it comes to the topic of risk. I believe the discussion and manuscript would benefit from tempering the language, as suicide risk scales and screening tools have not been the most reliable at predicting suicide. To ensure the results match the discussion, it would be good to ensure that the discussion is tempered and cautious, noting that dissociation may be a key factor and should warrant further investigation - but requires more study. 

3. Currently little detail is provided about the clinic. This may be done to preserve the identity of the participants. If so, please note that it is the case. Else, please provide more information on the clinic. 

4. Please provide more information on the area being studied, to help readers who are not familiar with Ankara, Turkiye understand the context and situation being faced by adolescents there. 

Thank you and I look forward to receiving a revised version of your manuscript.

All the best,

Dr. Sandersan Onie

---

## [Editor Report]

Dear Dr Açıkel, 

I am satisfied with your reviews, and am happy to recommend acceptance.

Thank you and all the best,

Sandersan Onie

---

## [Editor Report]

Dear authors,

Thank you for taking the time to revise the manuscript to this extent. I am recommending this for publication.

Thank you and all the best,

Sandersan Onie